# Biodiversity mediates ecosystem sensitivity to climate variability

Brunno F. Oliveira [1,2 ✉], Frances C. Moore [1] & Xiaoli Dong[1]

A rich body of evidence from local-scale experiments and observational studies has revealed stabilizing effects of biodiversity on ecosystem functioning. However, whether these effects emerge across entire regions and continents remains largely overlooked. Here we combine data on the distribution of more than 57,500 plant species and remote-sensing observations throughout the entire Western Hemisphere to investigate the role of multiple facets of plant diversity (species richness, phylogenetic diversity, and functional diversity) in mediating the sensitivity of ecosystems to climate variability at the regional-scale over the past 20 years. We show that, across multiple biomes, regions of greater plant diversity exhibit lower sensitivity (more stable over time) to temperature variability at the interannual and seasonal-scales. While these areas can display lower sensitivity to interannual variability in precipitation, they emerge as highly sensitive to precipitation seasonality. Conserving landscapes of greater diversity may help stabilize ecosystem functioning under climate change, possibly securing the continuous provisions of productivity-related ecosystem service to people.

[1] Environmental Science and Policy Department, University of California Davis, Davis, CA, USA. [2] Present address: Centre for the Synthesis and Analysis of Biodiversity (CESAB), FRB, Montpellier, France. ✉email: brunno.oliveira@me.com

Rapid global environmental change, including greater and more frequent extreme climate events, is profoundly transforming Earth's ecosystems[1–3]. Identifying factors that contribute to the persistence and stability of ecosystems despite these changes is fundamental for ensuring the continuous provision of services they underpin, such as carbon storage, timber, wildlife habitats and regulation of the hydrological cycle[4–6]. Biodiversity plays a critical role in stabilizing ecosystem functioning (i.e., less variable over time) as evidenced in a growing number of experimental[7–10] and observational[11–14] studies. Although fundamental to our basic understanding of ecosystem functioning and stability, this body of work has emphasized a limited range of ecosystem types (i.e., mostly grasslands) and smaller spatial scales (i.e., few square meters plots) than those relevant for management and policy[15–18]. Yet, sustaining ecosystem structure, functioning and services under future environmental conditions requires a deeper understanding of how biodiversity underpins their stability under realistic settings, across ecosystem types, and along climate gradients[19].

The magnitude and stability of many ecosystem processes, such as vegetation productivity and biomass production, are largely controlled by climate variability acting at multiple temporal scales[20,21]. This variability ranges from short-term climate extremes (e.g., heat waves) to seasonal climate dynamics that affect plant phenology, and to longer-term processes that can reflect climate change[22]. Recent global analyses using satellite remote-sensed data have revealed remarkable geographic variation in how vegetation production responds to fluctuation in multiple climate components over different temporal scales[20–24]. These patterns may at least in part reflect natural selection of a combination of successful plant life-history traits to patterns of climate variability in different regions[25,26]. For example, arid ecosystems (e.g., savannas and grasslands) show large amplitudes in vegetation green-ups to interannual variability in precipitation[24,27], while vegetation dynamics in seasonally cold ecosystems (e.g., temperate, arctic, and boreal biomes) are largely controlled by intra-annual variability in temperature, including climate-driven phenology[20,21]. Even while climate variability influences vegetation dynamics and idiosyncrasies of assemblage composition[26,28,29], it also reflects large-scale gradients of energy, resources, and other abiotic conditions that constrain, the diversity of plants, animals, and microbes, which is generally highest in temporally stable, warm and wet environments[30–32]. Yet, limited empirical evidence exists demonstrating that biodiversity can mediate the stability of ecosystems to climate variability at large spatiotemporal extents.

Much of the focus in biodiversity-stability studies has been built upon fine-scale and short-time periods in ecological systems. However, there is an increasing interest in coarse-scale (e.g., regional- or landscape-scale) perspectives of biodiversity–stability relationships to inform policy and conservation[15,33–36]. Landscapes of greater diversity are more likely to include a range of species that respond differently to environmental variation[37] and utilize different components of the resource base[38]. It is therefore expected that diverse landscapes would increase ecosystem stability via temporal and spatiotemporal niche partitioning[35,39–41]. This would occur because species may respond asynchronously to environmental fluctuation, such that variability of biomass production through time is reduced. Moreover, individuals of different species may occur in different vegetation patches across heterogeneous landscapes, and the resulting compositional turnover can increase stability due to spatial niche partitioning[17]. Even when a few highly productive species dominate biodiversity–stability relationships at the local-scale (i.e., selection effect[42]), differential responses to environmental changes by these few dominant species would reduce variability of biomass at the

landscape-scale, hence higher stability[39,40]. In contrast, inconsistent or even negative species richness-stability relationships may occur in regions where strong environmental filtering selects for a limited set of traits across different species (i.e., functional redundancy[43]) that respond synchronously to highly fluctuating resources[10], thus increasing temporal variability of vegetation productivity.

The current consensus is that biodiversity is much more than the simple sum of the species in a given locality. However, how multiple dimensions of biodiversity such as the richness of species (taxonomic diversity), the diversity of evolutionary lineages (phylogenetic diversity) and that of the traits related to ecological strategy (functional diversity) simultaneously influence ecosystem stability remains poorly investigated[7,44,45], in particular at large spatial scales. We explore the effects of multiple biodiversity dimensions on ecosystem stability, with a focus on those dimensions that are more directly related to hypothesized mechanisms, such as phylogenetic (PD) and functional diversity (FD) (Supplementary Fig. 1), which may provide deeper insights than focusing on species richness alone. While FD reflects how greatly assemblages differ in functional trait composition, PD is a metric of phylogenetic relatedness. Assuming phylogenetic relatedness reflects trait similarity, PD has been advocated as an indirect indicator of FD[46–48]. Yet, PD may capture a broader set of traits than is accounted for in FD measures[48], including hard-to-measuring traits[46,47]. Both PD and FD have shown stronger positive effects on ecosystem stability than richness in most studies[7,19,43,49–51], but not all[52]. PD and FD are more direct indicators for ecological differences among species than richness, hence niche-based processes. However, they may reveal little about spatiotemporal variability in the functional effects of species (i.e., temporal asynchrony and spatial complementarity). Therefore, by including species richness we expect to capture biodiversity–stability relationships related to the portfolio effect[38], when richness increases chances for asynchronous environmental responses by constituent species, regardless of their phylogenetic relatedness (i.e., PD) or trait similarity (i.e., FD).

In this study we set out to gain an understanding of the role of regional-scale biodiversity in modulating the co–variability between vegetation production and climate (e.g., ecosystem sensitivity[21]) across multiple terrestrial biomes, which to date is lacking. We assess this relationship using data on the species distribution, phylogeny and functional traits for more than 57,500 vascular plant species distributed across the Western Hemisphere (North, Central and South America)[53,54], at a 0.5° grid-cell resolution (~50 km$^2$ at the equator; see Methods). At each grid-cell ($N = 11,527$), we employ a multiple linear regression approach to identify regions (i.e., grid-cells) of strong vegetation sensitivity to climate variability, a key component of stability and resilience[21,55]. This study sheds light on how biodiversity can influence the sensitivity of ecosystem productivity to climate variability at the regional-scale, beyond the conventional plot-scales.

## Results and discussion

**Ecosystem sensitivity to climate variability.** The resilience of a system determines its capacity for absorbing changes to maintain fundamental controls on structure and function[55–57]. We estimate vegetation sensitivity to climate variability, a key component of resilience[56], by pairing 20 years' satellite-based time-series of vegetation productivity (enhanced vegetation index, EVI[58]) with that of two ecologically relevant climate components, temperature and precipitation[59]. These represent critical limiting factors for plant greenness and thus important drivers of vegetation

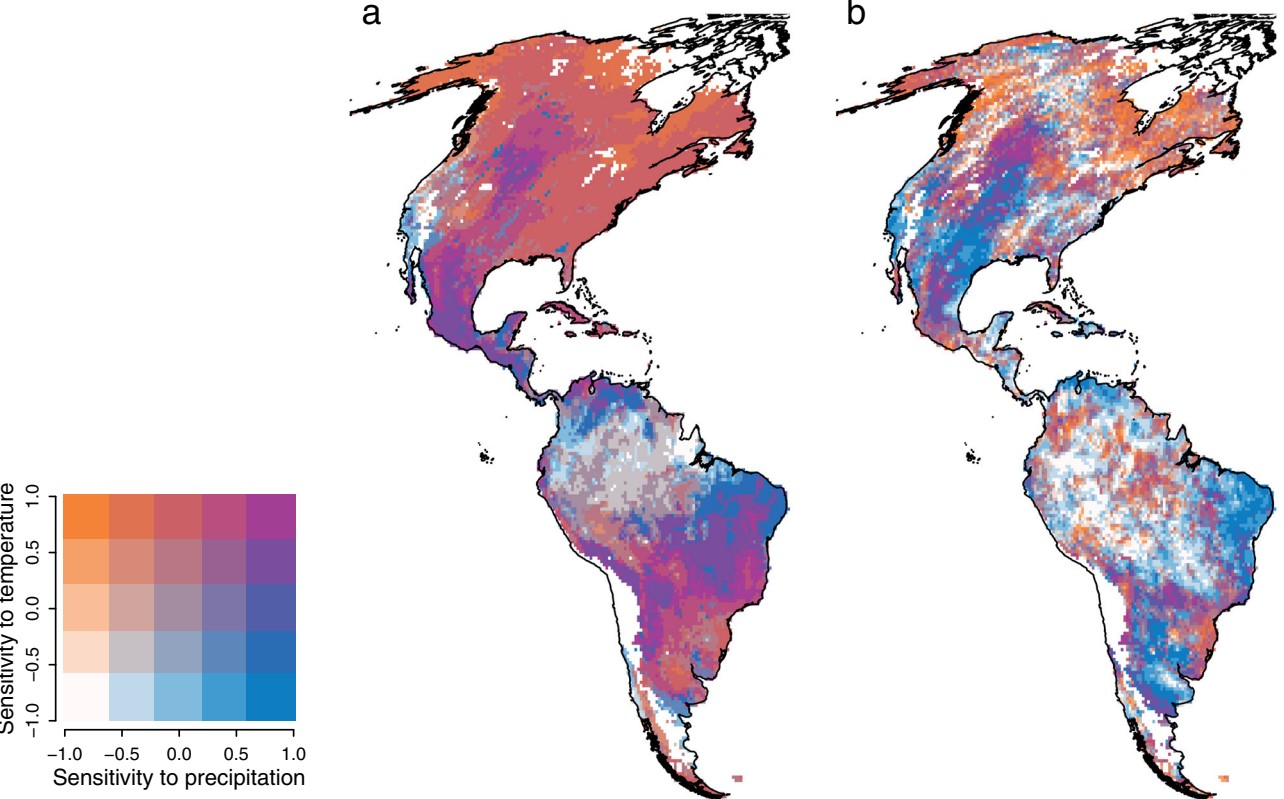

**Fig. 1 Geographic patterns of ecosystem sensitivity across the Western Hemisphere.** Bivariate plots representing ecosystem sensitivity to seasonality (**a**) and interannual changes (**b**) in temperature and precipitation. Ecosystem sensitivity is denoted by standardized coefficient from time-series of vegetation productivity against that of temperature and precipitation at each grid-cell (see methods). Sensitive ecosystems are those showing amplified response of vegetation production to climate variability. In contrast, ecosystems of low sensitivity are those of largely stable productivity conditions despite climate variability. White areas in the maps represent extremely sparse or inexistent vegetation cover and were eliminated from our analyses to reduce the potential impact of noisy data at low EVI values. For univariate geographical patterns see Extended Data Fig. 1.

temporal dynamics[20–22]. Given that vegetation carries the signature of climate variability at multiple time-scales[22], we explore ecosystem sensitivity to month-to-month (intra-annual) and year-to-year (interannual) oscillations in temperature and rainfall. While the former primarily captures short-term vegetation responses to climate variability (i.e., phenology dynamics, rapid gains in greenness shortly after rainfall, rapid drops in greenness that may reflect heat waves or insect outbreaks), the latter approximates mid-to-long-term vegetation responses to climate change (e.g., global warming and droughts). We refer to these temporal scales as seasonal and interannual, respectively.

Ecosystem sensitivity to temporal variability in temperature and precipitation varies systematically across the Western Hemisphere (Fig. 1, Supplementary Fig. 2, and Supplementary Fig. 3). Our framework depicts the typical phenology in vegetation greenness response to seasonality in temperature (Fig. 1a and Supplementary Fig. 2a), revealing a latitudinal gradient in the relative importance of temperature to vegetation dynamics (increasing temperature-limitation with latitude, Supplementary Fig. 4). At higher latitudes, temperature is also a limiting factor at the interannual-scale (Fig. 1b and Supplementary Fig. 2c), corroborating increasing evidence of greenness in polar areas in response to global warming[60,61]. The Amazon Forest shows relatively low-sensitivity levels, suggesting elevated resilience of tropical forests relative to most biomes in the Western Hemisphere (Fig. 1). In arid regions (e.g., South American Caatinga and Cerrado biomes, and North American Great Plains), vegetation shows positive sensitivity to seasonality in both temperature and precipitation, that is, more vegetation

greenness in warm and wet seasons (Fig. 1a, Supplementary Fig. 2b and Supplementary Fig. 3). Arid regions also show positive sensitivity to interannual variation in precipitation (Fig. 1b and Supplementary Fig. 2d). Thus, while vegetation greenness in these arid regions can respond rapidly to short-term fluctuations in temperature and precipitation, it is still highly limited by hydrologic stress from limited water supply in dry years. This agrees with previous findings highlighting plastic interannual vegetation dynamics in semi-arid ecosystems, which exert a strong influence on interannual variability of the terrestrial $CO_2$ sink[62,63]. The overall picture of our analysis on the vegetation sensitivity to climate variability is similar to previous modeling exercises conducted at either seasonal[20,21,23] or interannual-scales[24,27].

**Biodiversity effects on ecosystem sensitivity.** In order to control for the correlations among climate, productivity, and biodiversity, we investigate biodiversity–stability relationships within targeted ecosystem types that share similar climates and comparable distribution of life forms: biomes (e.g., shrublands, grasslands, forests), constituting historically and climatically well-defined bioregions[64]. Across multiple biomes, we show that plant diversity plays a significant role in affecting the spatiotemporal dynamics of ecosystem productivity (Fig. 2).

Plant richness and PD are the main drivers to reduce ecosystem sensitivity to temperature variability at both the interannual and seasonal-scales (Fig. 2a, c). PD has shown to stabilize biomass production over time at the plot-scale[45,65] and at the regional-scale[17], but to the best of our knowledge this is one

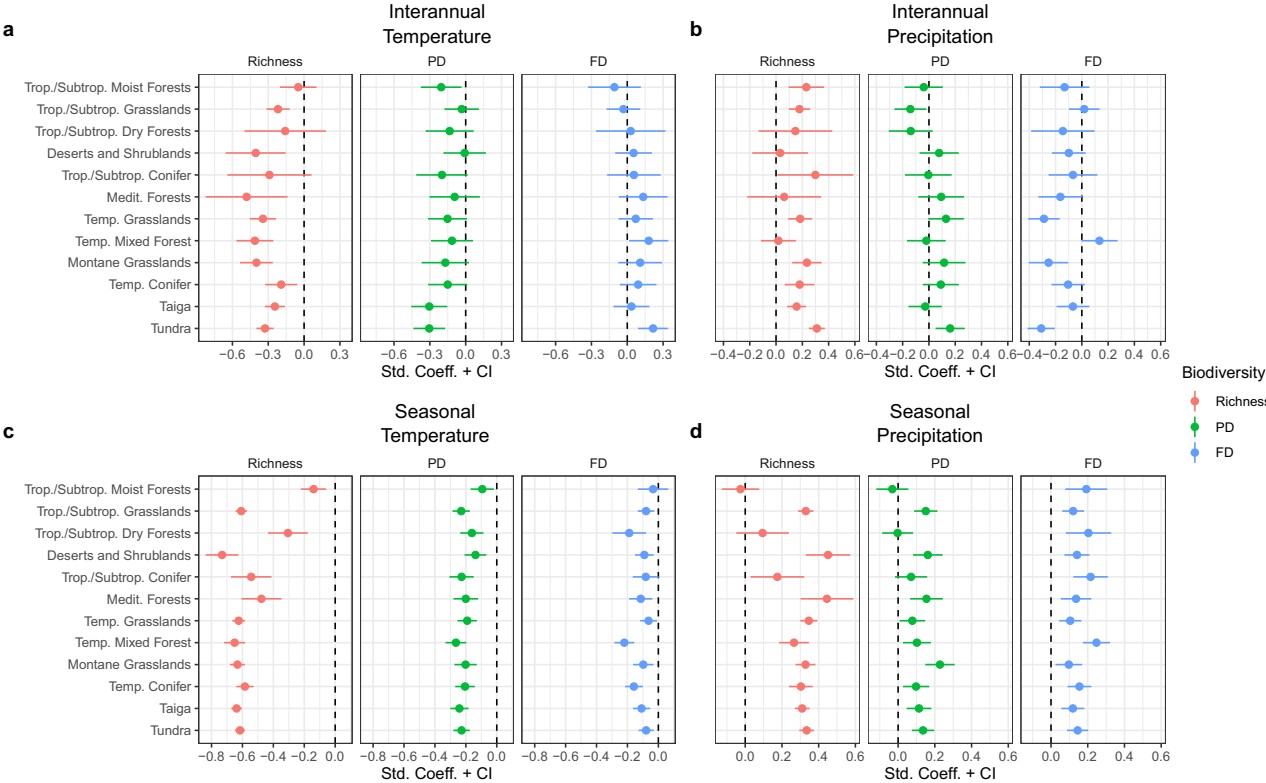

**Fig. 2 Biome-level biodiversity effects on ecosystem sensitivity.** Biodiversity effects on sensitivity to temperature (**a**, **c**) and precipitation (**b**, **d**) at the interannual (**a**, **b**) and seasonal (**c**, **d**) scales. Red, blue and green dots indicate richness, PD and FD effects, respectively. Dots and range lines represent standardized coefficients and 95% confidence intervals, respectively.

of the first studies to show that PD influences the sensitivity of vegetation production to climate variability. Assuming phylogenetic relatedness reflects ecological similarity, a high PD may increase spatiotemporal niche complementarity among distantly related species, hence buffering year-to-year climate variability[17,45,65]. Moreover, closely related species tend to share many traits, including pathogens or immune responses, via their shared co-evolutionary history[66]. As pathogens and herbivores often target a narrow range of phylogenetically related co-evolving species[66], the effects of pathogen outbreaks and herbivore attacks may be diluted in communities of high PD (e.g., spillover onto closely related species[66]). Therefore, if pests and pathogens outbreaks are triggered by abiotic conditions, landscapes dominated by phylogenetically clustered species would experience strong declines in biomass production as pests and pathogens spread across vegetation patches in response to environmental variability.

All biodiversity dimensions reduce ecosystem sensitivity to seasonal variability in temperature (Fig. 2c). These effects are stronger in colder biomes (Supplementary Fig. 5), where seasonality in temperature is highest. Landscapes of high biodiversity may increase resistance of biomass production to seasonality if they are composed of a set of species that respond asynchronously to environmental fluctuations. Moreover, a greater diversity of slow-growth species, capable of holding biomass despite seasonal climate fluctuation (e.g., via low leaf turnover), may prevent short-term (i.e., seasonal) climate variation from affecting baseline vegetation productivity levels. Indeed, using plant height and wood density as indicators of growth rate[67], we found that ecosystem sensitivity to temperature seasonality decreases in areas where trees are taller (Coefficient: −0.29, $p$-value: <0.001, $R^2$: 0.53) and wood density is higher (Coefficient: −2.98, $p$-value: <0.001, $R^2$: 0.44).

Sensitivity to temperature variability at the interannual-scale increases in areas of greater FD (Fig. 2a), with stronger effects in colder biomes (Supplementary Fig. 6). We identify increases in vegetation greenness in response to temperature trends over the last two decades (Supplementary Fig. 2c), consistent with previous research in cold biomes[60,61]. Our results add to these previous studies by suggesting that vegetation greenness response to global warming in cold environments is maximized in landscapes composed by diverse functional characteristics.

When assessing the contribution of biodiversity to ecosystem sensitivity to precipitation, our results reveal increases in sensitivity at both time-scales of climate variability in most biomes (Fig. 2b, d). An increase in ecosystem sensitivity may result from more fast-growing plant species, capable of increasing vegetation *responsiveness* to erratic resource availability shortly after rainfalls (e.g., nutrients induced by water availability), thus elevating rates of ecosystem recovery. This *responsiveness effect* is supported by all three biodiversity dimensions to short-term (i.e., seasonal) variability in rainfall (Fig. 2d). However, FD generally decreases sensitivity to interannual variability in precipitation (Fig. 2b), suggesting that functionally diverse landscapes can be more resistant, varying less in their productivity across dry and wet years.

Although our study demonstrates that PD and FD can influence ecosystem sensitivity in different biomes, the strongest and most consistent dimension of biodiversity contributing to ecosystem sensitivity is species richness (Figs. 2 and 3). As species richness increases at the regional-scale, it increases the likelihood of including a set of highly productive species in different vegetation patches and the chance for asynchronous population responses over time within each vegetation patch, both of which can potentially enhance the resistance of vegetation production to climate variability. Species richness can increase ecosystem

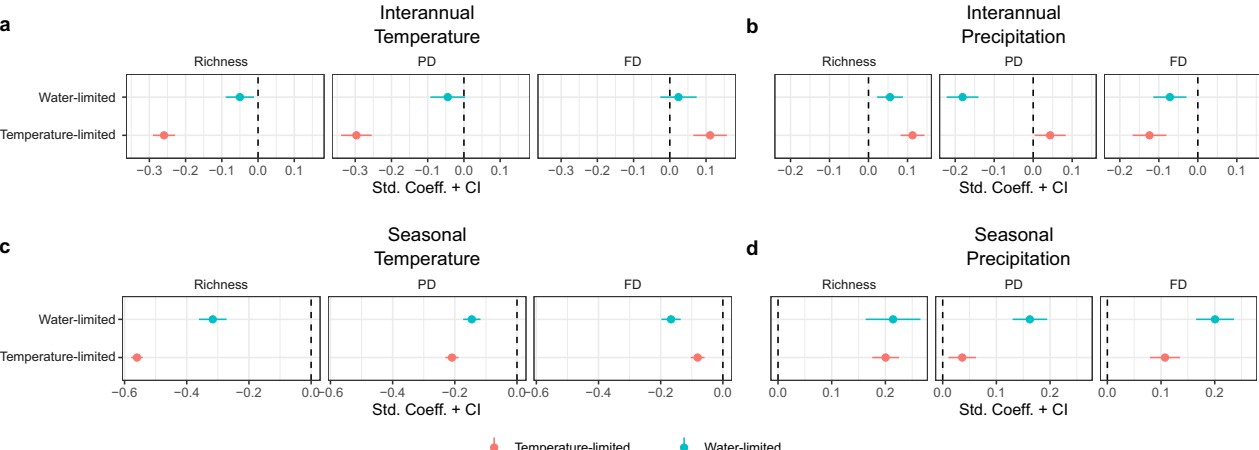

**Fig. 3 Biodiversity effects on ecosystem sensitivity in temperature-limited and water-limited regions.** Biodiversity effects on sensitivity to temperature (**a**, **c**) and precipitation (**b**, **d**) at the interannual (**a**, **b**) and seasonal (**c**, **d**) scales. Red and blue dots represent standardized coefficients of biodiversity (richness, PD and FD) in temperature-limited and water-limited regions, respectively. Range lines across dots represent 95% confidence intervals.

stability, regardless of levels of phylogenetic relatedness or ecological similarity among coexisting species, if it reflects the number of species with key functional traits that influence ecosystem functioning[68]. For example, high-biomass, slow-growth, long-lived, and deep-roots can increase the stability of biomass production over time[11,69,70]. Although fundamental for explaining many ecosystem processes[71], a high PD or FD may reflect an elevated diversity of lineages or traits that contribute little to vegetation productivity[68]. Identifying relevant functional traits and, perhaps more importantly, the functional composition that reflects structure and functioning of ecosystems in different biomes, should be a priority in research addressing questions related to ecosystem stability and management in a changing world[15,69].

To gain further insights into the role of biodiversity in stabilizing ecosystem productivity, we assess the extent to which the effect size of biodiversity changes in face of disturbances brought about by spatiotemporal variation in primary limiting factors on ecosystem productivity of a region. Variability in primary limiting factors of ecosystem productivity can generate amplified responses[20–22,24]. For example, regions of low water deficit show low sensitivity of vegetation productivity to precipitation variability because water is not the limiting factor (Fig. 1). We expect a strong effect of biodiversity in modifying the ecosystem sensitivity to the variability of the primary limiting factor of the region. This would occur if diverse landscapes are more efficient in their resource use, which enhances their productivity in conditions of limited resources[72,73]. Accordingly, we identified regions in which vegetation productivity is mainly limited by temperature and those by precipitation. We took the difference between ecosystem sensitivity to temperature and to precipitation for each temporal scale (seasonal and interannual), and classified temperature-limited regions as those with a stronger sensitivity to temperature than to precipitation, whereas water-limited were those with a stronger sensitivity to precipitation than to temperature (Supplementary Fig. 2).

Consistent with our expectations, we find strong effects of biodiversity in modifying ecosystem sensitivity to the variability in the primary regional limiting factor. For example, species richness and PD show stronger effects in reducing ecosystem sensitivity to temperature in temperature-limited regions than in water-limited regions (Fig. 3a, c), and FD enhances responsiveness to interannual variability in temperature more strongly in temperature-limited regions than in water-limited regions. PD and FD reduce ecosystem sensitivity to interannual variability in

precipitation in water-limited regions (Fig. 3b), while richness increases the responsiveness. Moreover, all biodiversity dimensions decrease ecosystem sensitivity to temperature seasonality in temperature-limited regions (Fig. 3c), whereas increase responsiveness to precipitation seasonality in water-limited regions (Fig. 3d).

Despite clear biogeographic congruences we identified between multiple biodiversity dimensions and ecosystem sensitivity metrics at the regional-scale, we acknowledge some limitations. First, biodiversity indices we used do not account for the evenness of biomass or abundance among species[38]. Second, these biodiversity indices do not account for temporal dynamics of production, which are more direct indicators of asynchrony[74]. Third, although species range maps often are well-suited for macroecological analyses and have been widely used[75], these inevitably depict imperfect spatial occurrence patterns (e.g., species ranges larger than those realized for some species). Nevertheless, the biogeographic pattern in plant richness we identify in our study is validated by an independent site-level plant occurrence database (SALVIAS), indicating a fairly accurate and robust description of regional-scale diversity patterns (See Supplementary Text, and Supplementary Fig. 7). Finally, we acknowledge our approach cannot definitively distinguish between correlation and causation because productivity and biodiversity covary with many environmental and historical drivers. Yet, the challenge in discerning causal relationships may be partly overcome by offering repeated evidence of patterns across different biomes. Results reported in our study call for future research to move from local-scale experiments and observations to broader-scale patterns to fully understand and predict the impacts of biodiversity change on ecosystem sensitivity to climate variability.

## Conclusion

Our study provides support for a mediating effect of regional-scale biodiversity patterns on the sensitivity of ecosystem productivity, and likely productivity-dependent ecosystem services, to temporal variability in environmental conditions. That biogeographical patterns of biodiversity largely coincide with those of low ecosystem sensitivity to temperature variability suggests sustaining biodiversity across the landscape may help stabilize ecosystem functioning under climate change. Nevertheless, vegetation productivity is driven by environmental conditions and species composition, both of which are affected by changing

climate[28,76,77]. As climate change alters the abundances and distributions of plant species[1,78–80], it also risks modifying ecosystem sensitivity to climate variability.

## Methods
All analyses were performed in R version 4.0.2[81].

**Environmental data.** We used the Enhanced Vegetation Index (EVI) derived from the Moderate Resolution Imaging Spectroradiometer (MODIS) sensor aboard the Terra Satellite as a proxy for vegetation productivity[58]. Vegetation productivity represents the total amount of carbon and energy fixed by plants that can be transfer up the food web to support the entire heterotrophic biomass in a given ecosystem[82], therefore being a key feature underpinning multiple ecosystem functions[83]. EVI is a normalized ratio of reflectance bands, with a practical range of 0 to 1, which has improved sensitivity over high-biomass regions than the other widely used Normalized Difference Vegetation Index (NDVI)[58]. We downloaded 20 years monthly aggregated EVI data from the MOD13C2 version 6 product at 0.05° (~5 km$^2$) resolution from the periods of February 2000 to December 2020 using the R package MODIS version 1.2.2. MOD13C2 represents cloud-free spatial composites achieved by replacing clouds with the historical MODIS time-series climatology record. We chose MODIS EVI over other vegetation products derived from sensors designed for vegetation monitoring because MODIS data is considered an improvement over other products[58].

We retrieved monthly data on temperature and precipitation from TerraClimate[59] that match the spatiotemporal resolution in the EVI data (i.e., 20 years' monthly values at a 0.05° spatial resolution). TerraClimate uses a spatial downscaling approach that employs bilinear interpolation of temporal anomalies to generate high-spatial resolution monthly climate datasets. Monthly temperature and precipitation products from TerraClimate were validated using station-based data obtained from the Global Historical Climatology Network (GHCN) database. This dataset shows noted improvement in overall mean absolute error and increased spatial realism relative to other available climate datasets[59].

**Biodiversity data.** We retrieved a collection of integrated and standardized data on vascular plants from the Western Hemisphere from the Botanical Information and Ecology Network (BIEN, http://bien.nceas.ucsb.edu/bien/) database version 4.2[53] using the R package BIEN version 1.2.4[54]. We started by selecting species with range maps and phylogenetic information as the absence of these data prohibit estimates of biodiversity dimensions (see below). For the purpose of our analyses, we decided for removing bryophytes ($N = 7691$) because their small biomass may contribute little to estimates of vegetation productivity, especially at large spatial scales. By keeping all other vascular plants we expect our data reflects the complex structural diversity that influence vegetation productivity patterns[84]. This resulted in 57,606 species for use in downstream analyses.

BIEN provides the largest set of standardized modeled range maps for plants available to date. Different range estimation methods were employed depending upon the sample size of (unique) presence records for each species. Most range maps were produced using species distribution models (SDM) generated in MaxEnt[85] with a combination of climate variables[86] and spatial eigenvectors[87], following recommendations for building less complex models[88]. For species with fewer than five occurrence records, a variety of approaches were employed as outlined in ref. [89]. Prior to species range estimation, occurrence records were cleaned and cultivated or nonnative occurrences removed. However, SDMs may still predict ranges larger than those realized for some species. To minimize this issue, cells where presence was predicted by MaxEnt further than 1000 km from any occurrence record were removed from the estimated range[89]. We must acknowledge that these range-map-derived data represent species' actual distributional patterns only at some relatively coarse resolution. Although the coarse resolution of our analyses (~50 km$^2$ grid-cells) has been considered well-suited for macroecological analyses[75], as any other species range prediction method, these predictions represent hypotheses about spatial occurrence patterns. An in-depth description of this modeling approach is available from ref. [89].

We downloaded 100 species-level multi-gene phylogenetic trees from BIEN. Genetic data used for estimating this phylogeny was queried from the GenBank based on the standardized list of Western Hemisphere species. Data extracted from GenBank was aligned using the standard single-run, unconstrained ML search method, and penalized likelihood was used to estimate divergence times from the molecular branch lengths of this tree. Taxa without genetic data were randomly grafted onto the base tree using taxonomy (genus membership) as a guide. Taxa not in the BIEN database were pruned from the ultrametric topology. Additional information on the BIEN phylogenies is available from ref. [54].

For traits, we used a combination of data retrieved from BIEN[54] and TRY[77] as to maximize trait coverage. Traits from TRY were requested from the web portal (https://www.try-db.org) and downloaded on 5/28/2021. We selected the 23 most complete and ecologically relevant traits (13 from BIEN and 21 from TRY). These traits are related to the global spectrum of plant form and function[67] (Supplementary Table 1), known to strongly influence the production and stability of plant biomass[71]. This resulted in more than 1 million trait values across species (Supplementary Table 1).

As many species lack trait information, we imputed missing values via a machine learning gap-filling algorithm with Bayesian Hierarchical Probabilistic Matrix Factorization (BHPMF[90]), which is a robust technic that has been used in other global trait-related studies[67,91–93]. The facts that many traits are strongly correlated and evolutionarily closely related species tend to occupy similar functional space suggest that imputation approaches must benefit from including allometric relationships among traits and evolutionary relationships among species. BHPMF accommodates both trait-trait correlation matrix and the phylogenetic trait signal via taxonomic hierarchy information[90]. To increase the imputation accuracy, we included phylogenetic information in the form of the phylogenetic eigenvectors (PEs[94],), as suggested by ref. [95]. PEs were calculated using the PVR package[96] in R. As PE calculation for large phylogenies is computationally prohibitive, we calculated PEs at the genus level ($N = 4498$ genera). Although we acknowledge that species-level PEs would be more informative than genus-level PEs, the fact that evolutionarily closely related species tend to be close in the functional space means that genus-level PEs must capture important aspects of phylogenetic relatedness. We randomly chose one species per genus, removed all other species, and computed PEs using the resulting genus-level tree. We repeated this process 100 times to capture phylogenetic uncertainty and averaged resulting PEs values across the 100 replicates. Across replicates, the first six PEs accounted for 43% of the variation of the whole phylogeny. We thus used the first six PEs in the final gap-filling process, excluding from the analyses low-representative PEs accounting for less than 3% phylogenetic variation[97]. We repeated the imputation approach for each one of the 100 phylogenies and averaged the resulting species-level imputed traits. We further used the minimum and maximum values per trait of the observed data as thresholds, replacing the gap-filled data with observed minimum or maximum when outside of the thresholds[92,93]. Finally, we selected eight ecologically relevant and commonly used traits[67,92,93] for further functional diversity analyses: leaf nitrogen content, leaf phosphorus content, leaf dry matter content, specific leaf area, leaf area, wood density, plant max height and seed dry mass. Imputations were performed using the R package BHPMF version 1.0.

Despite efforts in data collection, missing information is commonplace in life-history trait databases. This is a major challenge in macroecological studies, which often involve analyses across a large number of species (>10$^3$). A large percentage of missing data also occurs in our trait values to construct FD. Additionally, since phylogenetic and functional diversity are usually highly correlated[47,98], imputation of traits using the PEs could increase the intensity of the correlation between phylogenetic and functional diversity. Despite this common pitfall, model results show distinct effects of each biodiversity dimension on ecosystem sensitivity. Nonetheless, more data on the three dimensions of biodiversity are needed to improve our inferences. Regardless, the imputation approach for missing trait data we applied likely results in less bias than omitting data for species whose trait data were not available.

**Pre-processing.** We harmonized all variables with a Mollweide equal-area projection in a grid-cell resolution of 0.5° (~50 km$^2$). Any cell with a mean annual EVI below 0.1 was removed to reduce the potential impact of noisy data at low EVI values, which are attributed to areas with extremely sparse or inexistent vegetation cover[21]. We selected 13 well-established, geographically and climatically distinct biomes across the Western Hemisphere, as described by Olson et al.[64]. We excluded the "mangroves" due to small area ($N = 8$ grid-cells), and split "deserts and xeric scrublands" into "north desert" and "south desert" because the former gets seasonally colder than the latter. The biomes included in this study correspond to "tundra", "temperate conifers", "tropical and subtropical grasslands and savannas", "taiga", "flooded savannas", "montane savannas", "temperate mixed forest", "north desert", "Mediterranean forests", "subtropical conifers", "subtropical dry forests", "tropical forests" and "south desert".

**Ecosystem sensitivity metrics.** We used a multiple linear regression approach to estimate the relative influence of temporal viability in each climate variable in driving temporal changes in vegetation productivity. Ecosystem sensitivity was determined based on the standardized coefficients for temperature and precipitation extracted from multiple regressions fitted for each time-series. This approach has been used in previous conceptual modeling exercises looking at temporal vegetation responses to climate variability[20,21,23]. Prior to analyses, all time-series (for each grid-cell) were z-transformed using mean zero and unit variance to generate comparable (standardized) coefficients.

We estimated ecosystem sensitivity at two temporal scales: seasonal and interannual. While the former primarily captures short-term vegetation responses to climate variability (e.g., phenology dynamics, rapid drops in greenness that may reflect heat waves or insect outbreaks), the latter approximates mid-to-long-term vegetation responses to climate change (e.g., global warming and droughts)[22]. At the seasonal time-scale, monthly time-series of EVI and temperature, and precipitation were detrended using a time-series decomposition for removing distortions related to deterministic trends[99]. The function determines the trend component using a moving average, and removes it from the time-series. Therefore, our estimates of ecosystem sensitivity to seasonal variability in climate is unlikely to reflect deterministic trends. On the other hand, for estimates of ecosystem sensitivity to interannual climate variability, we aggregated monthly time-series to the annual-scale by calculating mean annual EVI and temperature,

and cumulative annual precipitation (following[86]). We did not detrend the interannual time-series as deterministic temporal increases or decreases are of interest at this scale.

Once we estimated ecosystem sensitivity, we identified regions in which vegetation productivity is mainly limited by temperature or precipitation. We referred to these regions as temperature-limited and water-limited regions, respectively. To this end, we classify grid-cells as temperature-limited if they show stronger sensitivity to temperature than to precipitation. Likewise, we classify grid-cells as water-limited if they show stronger sensitivity to precipitation than to temperature (Supplementary Fig. 2). As sensitivity to temperature is stronger in temperature-limited ecosystems, we expect that biodiversity would lower sensitivity to temperature to a greater degree in these regions than it does in other regions. Likewise, as ecosystem sensitivity to precipitation is stronger in water-limited regions, we expect that biodiversity would lower sensitivity to precipitation in these regions to a greater degree compared to its effect in other regions.

**Biodiversity dimensions metrics**. A myriad of approaches has been developed to measure biodiversity and its multiple dimensions (i.e., species richness, PD and FD). We chose biodiversity metrics sharing similar mathematical proprieties that have been classified as richness metrics[100,101]. Biodiversity dimensions were calculated for each 0.5° grid-cell.

We determined vascular plant richness by overlaying species range maps and counting the number of species that overlap at each grid-cell. PD was estimated with the widely used Faith's index[48], which quantifies the amount of evolutionary history of a set of species in terms of millions of years. PD sums across all the branch lengths connecting species in a phylogeny, from tips to root. PD was calculated on a maximum clade credibility (MCC) tree estimated from the 100 original BIEN trees using the 'maxCladeCred' function from the "phangorn" R package[102]. For FD, we used functional richness, representing the volume of the trait space encapsulated by a group of species (e.g., convex hull[103]). We summarized the trait space using a principal coordinate analysis (PCA) carried out on the eight selected traits. A single trait space including all species was used to preserve total inertia and distance between the same species occurring in different assemblages[104]. We selected the first three PCA axes as traits for inferring FD as these captured >80% of the total variance in traits. Functional richness was calculated in R using the FD package[105]. We avoided the use of dendrogram-based FD indices since recent studies showed that these indices may lead to biased estimates and inaccurate ecological conclusions[104,106,107].

Owing to the mathematical proprieties of these metrics, PD and FD values can be highly dependent on species richness (Supplementary Fig. 8). We remove this artifact by quantifying the amount of deviation by the observed PD or FD from the null expectation given the species richness observed in a grid-cell. This was done by applying a null model that kept grid-cell richness constant while randomizing assemblage composition, and recalculating PD and FD[108,109]. This procedure was repeated 1000 times to generate a null distribution of PD and FD values, which was contrasted against the observed PD and FD values using the formula of standardized effect size (SES) [SES = (observed − mean(null))/sd(null)]. Positive SES indicates assemblages with more PD or FD than expected by richness, while negative SES indicates assemblages with more PD or FD than expected by richness.

**Statistics and reproducibility**. We used spatial simultaneous autoregressive (SAR) models with a spatial error term to analyze the data while accounting for spatial autocorrelation. SAR models incorporate spatial autocorrelation with weight matrices that specify the strength of interaction between neighboring sites[110,111]. We assessed the effect of multiple weighting functions when defining the matrix of spatial weights and chose the one that best accommodated the spatial structure present in the variables. We built a connectivity matrix defined by the distance in which Moran's $I$ was strongest, and an inverse distance weighting function ($1/d$) was necessary to account for spatial structure.

We fitted SAR models for each vegetation sensitivity metric (temperature and precipitation) and at the two temporal scales (seasonal and interannual). We used a weighting scheme for accommodating uncertainty in ecosystem sensitivities estimates. To this end, we used the inverse of the range in confidence intervals from each sensitivity metric as weights in our SAR models. Our model structure includes as predictor variables the three biodiversity dimensions, biome, regional limitation factors (water-limited vs. temperature-limited), interactions between each biodiversity dimension and biomes, and interactions between each biodiversity dimension and regional limiting factor. We model each vegetation sensitivity metric as absolute values, as values close to zero are assumed to be more stable. All variables were z-transformed to a mean of zero and unit variance to make model coefficients comparable. SAR models were fitted in R, using the R package spatialreg[112].

**Reporting summary**. Further information on research design is available in the Nature Research Reporting Summary linked to this article.

## Data availability

The project was based entirely on data that are publicly available through BIEN (https://bien.nceas.ucsb.edu/bien/), TRY (https://www.try-db.org/), WWF biomes (https://www.

worldwildlife.org/publications/terrestrial-ecoregions-of-the-world), NASA MODIS vegetation index (https://lpdaac.usgs.gov/products/mod13c2v006/), and TerraClimate (https://www.climatologylab.org/terraclimate.html).

## Code availability

Code used for generating all analyses can be found at: https://github.com/oliveirab/EcoSens.

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

## Acknowledgements

We are thankful to Susan P. Harrison for her valuable comments on an early version of this study. Thanks to BIEN and TRY for collating and maintaining the traits and range data used in this study. This study was supported by the National Science Foundation (award number 1924378: "CNH2-S: Understanding the Coupling Between Climate Policy and Ecosystem Change").

## Author contributions

All authors conceived the study. B.F.O. ran analyses and wrote the manuscript with input from F.C.M. and X.D.

## Competing interests
The authors declare no competing interests.
