## [Peer Review File · Communications Biology]

nature portfolio

Peer Review File

Biodiversity mediates ecosystem sensitivity to climate variabilityPeer Review Information

Journal: Communications Biology

Manuscript Title: Biodiversity mediates ecosystem sensitivity to climate variability

Corresponding author name(s): Brunno F. Oliveira

Editorial Notes:

Transferred manuscripts This manuscript has been previously reviewed at another *Nature Research* journal. This document only contains reviewer comments and letters for versions considered at *Communications Biology*.

Reviewer Comments & Decisions:

Reviewers Comments:

Reviewer #1 (Remarks to the Author):

[REDACTED]

Biodiversity mediates ecosystem sensitivity to climate variability

[REDACTED] I do not feel the need to re-iterate the main findings. Overall, I have a much better understanding of what the authors have done, the methods are suitably clear and helpful caveats have been included. I do agree that the findings they have made are well justified and suitably novel for publication.

I am happy for this manuscript to be published with just a few minor corrections:

We are glad your revised manuscript satisfies the Reviewer. We have addressed all the corrections suggested by the Reviewer. Thank you for highlighting these issues.

1. The captions for Supplementary figures 5 and 6 may be switched or the main text is referring to the wrong one, as Supp Fig 5 seems to be for interannual-scale changes, but it is referred to the in the text (Line 163) as illustrating seasonal relationships.

We fixed Supplementary figure numbers.

2. Line 326 refers to a table in supplementary materials, but there is not one.

We added the Supplementary table.

3. Line 441 refers to supplementary figure 2 as illustrating a relationship between richness, PD

and FD, but that figure is the map of ecosystem sensitivities at different temporal scales and climate variables.

We added a new Supplementary figure (Supplementary Fig. 8) which illustrates pairwise relationships among biodiversity dimensions.